# Ada-MoGE: Adaptive Mixture of Gaussian Expert Model for Time Series Forecasting

## Abstract

Multivariate time series forecasts are widely used, such as industrial, transportation and financial forecasts. However, the dominant frequencies in time series may shift with the evolving spectral distribution of the data. Traditional Mixture of Experts (MoE) models, which employ a fixed number of experts, struggle to adapt to these changes, resulting in frequency coverage imbalance issue. Specifically, too few experts can lead to the overlooking of critical information, while too many can introduce noise. To this end, we propose Ada-MoGE, an adaptive Gaussian Mixture of Experts model. Ada-MoGE integrates spectral intensity and frequency response to adaptively determine the number of experts, ensuring alignment with the input data's frequency distribution. This approach prevents both information loss due to an insufficient number of experts and noise contamination from an excess of experts. Additionally, to prevent noise introduction from direct band truncation, we employ Gaussian band-pass filtering to smoothly decompose the frequency domain features, further optimizing the feature representation. The experimental results show that our model achieves state-of-the-art performance on six public benchmarks with only 0.2 million parameters.

## 1 Introduction

Time series forecasting models hold immense application value and are widely utilized in fields such as industrial manufacturing, finance, and meteorology. Historically, time series forecasting models have primarily been categorized into four architectural families: RNN Hochreiter & Schmidhuber (1997), MLP Zeng et al. (2022), Transformer Vaswani et al. (2023), and Mamba Gu & Dao (2024). In recent years, Mixture of Experts (MoE) models have gained popularity in large language models due to their diverse feature representations and accelerated inference capabilities. The inherent characteristic of mixture of experts models, where different experts handle different features, naturally lends itself to processing features of varying frequencies in time series forecasting tasks. Frequency domain features represent the periodicity of signals, and capturing complex periodic signals is crucial for time series forecasting tasks.Therefore, the application of mixture of experts in time series forecasting tasks is urgently in need of exploration.

Recently, several hybrid expert-based time series forecasting models have been proposed. Time-MoE Liu et al. (2025) constructs a MoE model with 2.4 billion parameters, replacing the feed-forward layers in the transformer with MoE layers. However, the model primarily focuses on time-domain features while neglecting the importance of frequency-domain features. MoFE-Time Liu et al. (2024c) equips every expert with parallel FFT MLP and time domain MLP branches. However, it merely applies the expert to the whole spectrum without decoupling individual frequencies, so dominant harmonics stay mixed with noisy bins and are easily missed. FreqMoE Yang et al. (2025) takes a step further by splitting the spectrum into sub-bands fusing each expert based on a soft-weighting approach. However, this soft-weighting method does not explicitly discard noise experts, making it unable to completely filter out noisy frequency bands, which results in suboptimal performance.

Furthermore, employing the Hard MoE method that retains the top K experts in the frequency domain also presents issues. The number of experts in a Hard MoE model is fixed, which can lead to an imbalance in frequency coverage. The range of the dominant frequency domain varies for different data, thus requiring a different number of experts. Selecting fewer experts may result in the omission of major frequencies. On the other hand, selecting too many experts may introduce noise bands,

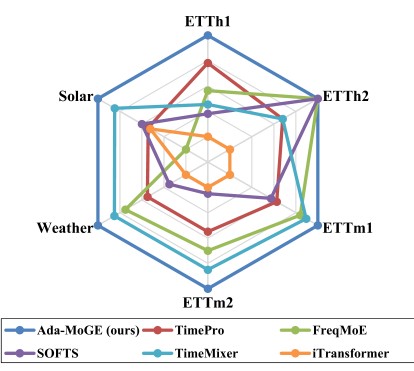

(a) The Performance of Ada-MoGE

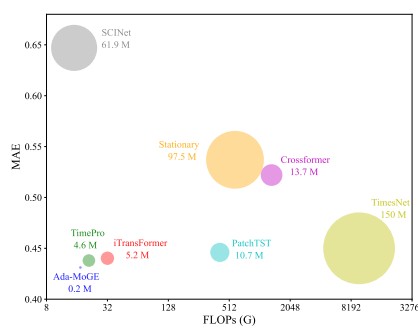

(b) Comparison of Parameters and FLOPs

Figure 1: Performance comparison of Ada-MoGE with other state-of-the-art Models. Figure (a) shows a radar map based on MSE which shows that AdaMoGE has achieved advanced performance on six public benchmarks. Figure (b) shows the parameters and FLOPs of Ada-MoGE versus other state-of-the-art models. The parameter of Ada-MoGE is only 0.2M, and the FLOPs are significantly less than those of the existing models. And the MAE on ETTh1 of our model is significantly lower than that of other models.

causing the dominant frequencies to be drowned out by noise. This frequency coverage imbalance issue limits the performance of frequency domain MoE models in time series forecasting.

To address the frequency coverage imbalance issue, we propose an adaptive mixture of Gaussians experts model, named Ada-MoGE, which can adaptively select the number of experts based on the input data. It simultaneously computes univariate spectral intensity and cross-variable frequency response features for fusion, and uses the fused features to adaptively determine the number of activated experts. The joint representation of frequency and variable dimensions enables the model to learn high-energy regions representing dominant frequencies and high-energy channels representing sensitive variables, thereby activating only the experts that process dominant frequencies as much as possible. Experts with higher noise levels are explicitly turned off to reduce noise interference. This approach effectively resolves the frequency coverage imbalance caused by processing different data.

Besides, to reduce the introduction of time-domain noise caused by direct truncation in the frequency domain, we designed Gaussian experts to perform soft decoupling of the frequency-domain features. Specifically, we first pass the input sequence through a set of learnable Gaussian band-pass filters whose center frequencies are optimized end-to-end. Each resulting sub-band is assigned to a lightweight expert network. This design ensures that the true dominant frequency becomes the principal component of at least one expert's input, eliminating cross-band interference and allowing each expert to capture fine-grained, frequency-specific dynamics without disturbance. To this end, the proposed Ada-MoGE can achieve less noise introduction and more comprehensive retention of dominant frequency features. The experimental results in Figure 1 demonstrate that our model has achieved state-of-the-art performance on six benchmark datasets, with only 0.2M parameters and significantly lower FLOPs compared to existing methods.

## 2 RELATED WORK

### 2.1 TIME SERIES FORECASTING METHOD

Current advanced time series forecasting primarily includes linear models, Transformer, Mmaba, and other architectures. Firstly, linear models represented by DLinear Zeng et al. (2022), and RLinear Kim et al. (2023) directly perform regression on historical sequences using a single layer or very shallow MLP. They have achieved advanced performance on long sequence benchmarks due to their small parameter size, fast training, and stability over long windows. Secondly, Transformers capture dependencies of arbitrary distances through self-attention or sparse attention. Models like PatchTST Nie et al. (2023), FEDformer Zhou et al. (2022), and iTransformer Liu et al. (2024b), continuously reconstruct normalization, patch division, and frequency domain attention, maintaining

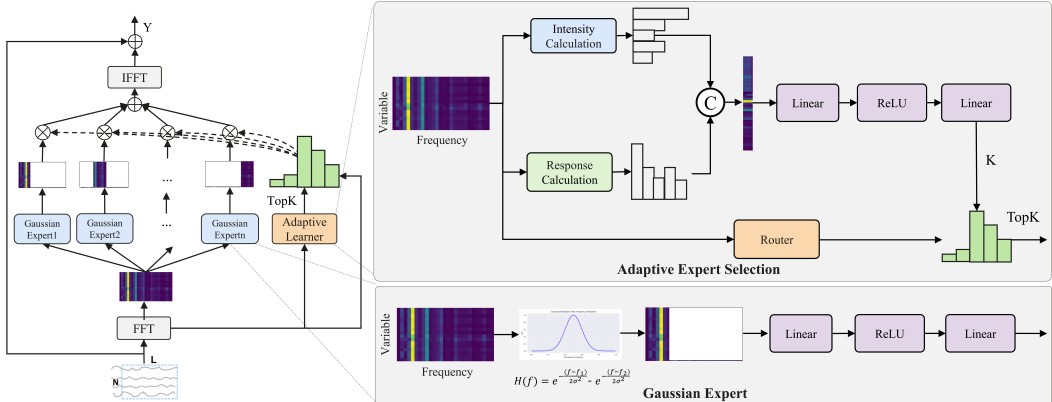

Figure 2: The overview of the Ada-MoGE method. It integrates spectral intensity and frequency response to adaptively determine the number of experts, thereby ensuring that the number of experts matches the frequency distribution of the input data. Additionally, to prevent noise introduced by direct band truncation, experts based on Gaussian band-pass filters are employed to smoothly decompose the frequency-domain features.

leading accuracy in multivariate and multi-step prediction tasks. However, their quadratic complexity and memory consumption remain bottlenecks for practical deployment. In recent years, mamba-based methods represented by S-Mamba Wang et al. (2025) and TimePro Ma et al. (2025) have begun to emerge, reducing complexity to O(L) while retaining global receptive fields, providing new scalable solutions for long-range time series modeling. However, existing methods generally rely on end-to-end training for feature extraction, failing to explicitly decouple dominant frequency components, which results in critical spectral information being overwhelmed by redundant features, thus becoming a bottleneck for further performance improvement.

## 2.2 Mixture of Experts in Time Series Forecasting

In recent years, the Mixture-of-Experts (MoE) paradigm has begun to migrate from the fields of NLP and CV to the domain of time series prediction, aiming to expand model capacity while maintaining low inference costs. Time-MoE Liu et al. (2025) is the first to replace the expert layer in the Transformer's Feed-Forward layer, allowing different experts to capture distinct features. However, it does not practically assign different inputs to each expert. MoFE-Time Liu et al. (2024c) equips every expert with parallel FFT MLP and time domain MLP branches, using sparse MoE routing to assign frequency components to the most appropriate expert. However, it merely applies the MLP to the whole spectrum without decoupling individual frequencies, so dominant harmonics stay mixed with noisy bins and are easily missed. FreqMoE Yang et al. (2025) takes a step further by splitting the spectrum into sub-bands and routing each to a dedicated expert before SoftMoE fusion, yet the abrupt band-wise truncation introduces Gibbs-like edge oscillations when the signal is converted back to the time domain, inadvertently re-introducing interference. In summary, existing MoE models are not well-equipped to effectively address the dominant frequency suppression issue.

## 3 Method

### 3.1 Overview of Ada-MoGE

In time series forecasting, the dominant frequency may vary according to changes in the frequency distribution of the data. Using a fixed number of experts often leads to an imbalance in frequency domain coverage. To address this issue, we propose an adaptive Gaussian Mixture-of-Expert model called Ada-MoGE. It integrates spectral intensity and frequency response to adaptively determine the number of experts, ensuring that the number of experts matches the frequency distribution of the input data. This approach avoids information loss due to too few experts and noise introduction due to too many experts. Additionally, to prevent noise introduction caused by direct truncation of frequency bands, we employ Gaussian band-pass filtering to smoothly decompose the frequency

domain features, further optimizing the feature representation. Moreover, our Ada-MoGE is highly flexible. It can replace the FFN layers in Mamba and Transformer models or be used independently.

## 3.2 Adaptive Expert Selection

To adaptively select the expert that dominates the frequency and suppresses noise, we propose an adaptive expert selection mechanism. It captures both the dominant frequency and key features by simultaneously capturing the average spectral intensity and the cross-variable average frequency response, achieving a dual-drive frequency-variable adaptive expert selection.

First, the Fast Fourier Transform (FFT) is employed to convert time-domain features into the frequency domain. Subsequently, to identify the dominant frequencies, we first perform cross-variable averaging at each frequency. Specifically, we sum the magnitudes of the Fourier coefficients across all $V$ channels at the same frequency $f$, and then divide by $V$ to obtain the cross-variable averaged frequency response $\mu(f)$. This $\mu(f)$ reflects the overall response intensity of the system at each frequency point, aiding in the identification of the system's dominant frequencies. When $\mu(f)$ at a certain frequency is significantly higher than its neighboring frequencies, it indicates that the frequency component is highly reproducible across different variables, likely corresponding to the system's inherent period or external forcing signal. Conversely, if $\mu(f)$ is at a low level without significant peaks, it suggests that the frequency is dispersed across channels, with the energy source primarily being random noise or measurement error, thus having limited value for subsequent predictions. Therefore, $\mu(f)$ provides a global clue as to which frequencies are truly important. The formula for calculating the cross-variable averaged frequency response is as follows:

$$\mu(f) = \frac{1}{V} \sum_{v=1}^{V} |X_v(f)|, \tag{1}$$

where $V$ denotes the number of variables and $X_v(f)$ the complex FFT result of variable $v$ at frequency $f$. The vector $\boldsymbol{\mu} \in \mathbb{R}^F$ will be referred to as the frequency response vector.

Furthermore, to identify key variables, the average spectral intensity $E(v)$ for each variable is calculated. Specifically, we sum the amplitudes of all frequencies for the variable $v$ and then divide by the maximum frequency $F$. $E(v)$ reflects the average intensity of the variable across the entire frequency domain and can serve as a quick indicator of its activity level. A larger $E(v)$ typically indicates the presence of significant seasonal components or high-frequency switching in the variable. Conversely, a smaller $E(v)$ suggests that the sequence tends to be stationary or subject to strong damping. The formula for calculating the cross-variable average frequency response across the entire band is as follows:

$$E_v = \frac{1}{F} \sum_{f=0}^{F-1} |X_v(f)|, \tag{2}$$

where $F$ represents the number of frequencies, $X_v(f)$ denotes the complex Fast Fourier Transform (FFT) result of variable $v$ at frequency $f$. The vector $E_v$ will be referred to as the spectral intensity vector.

By concatenating $E(v)$ with $\mu(f)$, the model can simultaneously grasp two complementary pieces of information: the dominance in the frequency dimension and the activity in the variable dimension. The former informs the model which frequencies to focus on, while the latter indicates which variables to pay attention to, enabling the model to accurately capture key features for learning the number of experts. The fused feature vector is further refined by an MLP to output the selected number of experts K to a lightweight gating network. The gating network outputs the activation probabilities of the experts and selects the top K experts for activation. In this way, an adaptive expert budget allocation driven by both frequency and variable is achieved, enhancing the ability to capture dominant frequencies. The overall formula is as follows:

$$\mathbf{K} = \mathbf{W}_2 \cdot \sigma(\mathbf{W}_1 \cdot \boldsymbol{\chi}(\boldsymbol{\mu}, \mathbf{E}) + \mathbf{b}_1) + \mathbf{b}_2 \tag{3}$$

where $\boldsymbol{\chi}(\boldsymbol{\mu}, \mathbf{E})$ denotes the concatenation of the frequency response vector $\boldsymbol{\mu} \in \mathbb{R}^F$ and the spectral intensity vector $\mathbf{E} \in \mathbb{R}^V$. $\mathbf{W}_1 \in \mathbb{R}^{H \times (F+V)}$ and $\mathbf{W}_2 \in \mathbb{R}^{D \times H}$ are the weight matrix of the linear layer. $\mathbf{b}_1 \in \mathbb{R}^H$ and $\mathbf{b}_2 \in \mathbb{R}^D$ are bias vectors. $\sigma(\cdot)$ is the ReLU activation function.

### 3.3 GAUSSIAN FEATURE DECOUPLING

To avoid the noise introduced by direct band truncation, we employed a Gaussian band-pass filter to perform a smooth decomposition of the frequency domain features. Specifically, based on the frequency domain features obtained from the fast Fourier transform, a Gaussian band-pass filter was established for feature filtering. The Gaussian band-pass filter retains only the energy within the target passband while exponentially suppressing the out-of-band components, ensuring a smooth filtering characteristic. The boundaries of each Gaussian band-pass filter are learned through an end-to-end optimization process. This optimization process automatically reallocates the support regions of the filters. Information-rich frequency bands are directed to experts with higher activation values, while frequency bands dominated by interference or noise are directed to other experts with lower activation values. This learnable frequency partitioning strategy avoids spectral aliasing, providing each expert with a statistically independent input subspace and laying the foundation for the adaptive selection of dominant frequency bands. The equation for Gaussian band-pass filtering is as follows:

$$H(f) = \exp\left(-\frac{(f - f_1)^2}{2\sigma^2}\right) - \exp\left(-\frac{(f - f_2)^2}{2\sigma^2}\right) \tag{4}$$

where $H(f)$ denotes the frequency response of the filter at frequency $f$. The parameters $f_1$ and $f_2$ represent the upper and lower cutoff frequencies of the passband, respectively. The term $\sigma$ is the standard deviation that controls the bandwidth of each Gaussian component. A larger $\sigma$ results in a smoother transition and wider frequency coverage. By subtracting two Gaussian functions centered at $f_1$ and $f_2$, this formulation creates a bandpass effect that suppresses both low and high frequencies while preserving those within the desired range.

Besides, we design a spectrum-driven adaptive standard deviation mechanism. This method automatically determines the standard deviation $\sigma$ by analyzing the energy distribution of the frequency spectrum. Specifically, we first compute the average spectral intensity and normalize it by the current center frequency. As the center frequency increases, the standard deviation decreases dynamically, achieving adaptive frequency tuning. The low-frequency band often contains the long-term periodic components of a signal, such as seasonal variations or long-term trends. A larger standard deviation can better capture these components, ensuring that the filter does not miss important periodic information. The high-frequency band often contains more noise components. A narrower filter bandwidth can more accurately separate the useful components in the signal, avoiding the misjudgment of noise as signal. Meanwhile, we employ a parameter $\alpha$ to adjust its magnitude, ensuring it remains within a reasonable range. Compared to manually setting $\sigma$, this method is more adaptable to different frequency band characteristics.

$$\sigma_j = \sigma_0 \cdot \frac{\alpha}{D_j} \cdot \frac{1}{N} \sum_{i=1}^{N} |X(f_i)|^2 \tag{5}$$

where the $\sigma_j$ denotes the automatically determined standard deviation, which controls the bandwidth of the Gaussian bandpass filter. $\sigma_0$ is the initial standard deviation. The term $D_j$ represents the center frequency of the filter. $\alpha$ refers to the adjustment coefficient.

## 4 EXPERIMENT

### 4.1 DATASETS

We evaluate our method on widely-used benchmarks for long-term multivariate time-series forecasting, covering electricity load, renewable energy, and meteorology. ETTh1/ETTh2 Zhou et al. (2021) are two hourly datasets originate from transformer load and oil temperature monitoring. ETTm1/ETTm2 Zhou et al. (2021) are the minute-level counterparts of ETTh1/ETTh2, sampled every 15 minutes. ECL (Electricity Consuming Load) Wu et al. (2021) is a time series data set of power loads, which records hourly consumption from 321 clients. The Weather Angryk et al. (2020) is a real-world weather dataset. Solar-Energy Lai et al. (2018) is collected from 137 solar power plants, which provides 10-minute resolution energy production data. Unless specified otherwise, all datasets follow the standard train/validation/test splits with prediction horizons of $\{96, 192, 336, 720\}$.

Table 1: Performance comparison of models before and after integrating the Ada-MoGE module.

| Models | Metric | TimeMixer MSE | TimeMixer MAE | TimePro MSE | TimePro MAE | iTransformer MSE | iTransformer MAE | PatchTST MSE | PatchTST MAE |
|--------|--------|-----|-----|-----|-----|-----|-----|-----|-----|
| ETTh1 | Original | 0.447 | 0.440 | 0.438 | 0.438 | 0.454 | 0.447 | 0.453 | 0.446 |
|       | +Ada-MoGE | **0.432** | **0.431** | **0.436** | **0.433** | **0.453** | **0.447** | **0.438** | **0.434** |
| ETTh2 | Original | 0.383 | 0.407 | 0.377 | 0.403 | 0.383 | 0.407 | 0.385 | 0.410 |
|       | +Ada-MoGE | **0.373** | **0.400** | **0.374** | **0.401** | **0.378** | **0.403** | **0.371** | **0.399** |
| ETTm1 | Original | 0.381 | 0.395 | 0.391 | 0.400 | 0.407 | 0.410 | 0.396 | 0.406 |
|       | +Ada-MoGE | **0.377** | **0.395** | **0.386** | **0.396** | **0.406** | **0.408** | **0.384** | **0.398** |
| ETTm2 | Original | 0.275 | 0.323 | 0.281 | 0.326 | 0.288 | 0.332 | 0.287 | 0.330 |
|       | +Ada-MoGE | **0.272** | **0.321** | **0.276** | **0.321** | **0.286** | **0.330** | **0.281** | **0.327** |

## 4.2 IMPLEMENTATION DETAILS

**Optimization and Metrics**    The models are trained with mean squared error (MSE) as the objective. During evaluation, both MSE and mean absolute error (MAE) are reported to reflect variance- and bias-related performance. We adopt the Adam optimizer combined with cosine annealing for gradual learning rate decay.

**Model and Hardware Configuration**    We conduct a grid search over the following hyperparameters: the maximum number of experts $\in \{5, 6, 7, 8, 9, 10\}$, the encoder depth $\in \{1, 2, 3, 4\}$, and the feature dimension $\in \{8, 16, 32\}$. All experiments were run on eight NVIDIA Tesla V100 GPUs.

## 4.3 MAIN RESULTS

To evaluate the effectiveness of the proposed Ada-MoGE module, we integrate it into several existing models. As shown in Table 1, the integration yields consistent performance improvements by a clear reduction in both MSE and MAE metrics in most cases. For instance, on the ETTh2 dataset, PatchTST with Ada-MoGE achieves an MSE of 0.371 and MAE of 0.399, outperforming its original scores of 0.385 and 0.410. Similarly, for TimeMixer on the ETTm1 dataset, the MSE decreases from 0.391 to 0.386 after integration. It is noteworthy that in the few scenarios where significant gains are not observed (e.g., iTransformer on ETTh1), the performance remains on par with the original model, indicating that the module introduces no detriment. Among all enhanced models, Ada-MoGE empowers TimeMixer to achieve the most competitive performance.

Table 2 presents a performance comparison of various state-of-the-art time series forecasting models, including Ada-MoGE (our proposed model), TimePro Ma et al. (2025), FreqMoE Yang et al. (2025), SOFTS Han et al. (2024), TimeMixer Liu et al. (2024a), iTransformer Liu et al. (2024b), PatchTST Nie et al. (2023), and TimesNet Wu et al. (2023), across several datasets, with different forecasting horizons (96, 192, 336, and 720) and a fix lookback window 96. Ada-MoGE consistently achieves the best performance in terms of both MSE and MAE across multiple datasets and time horizons. For example, in the ETTh1 dataset, Ada-MoGE delivers the best MAE of 0.388 at the 96-step horizon, outperforming TimePro (0.394) and FreqMoE (0.399). This trend of outperforming other models is also observed across other datasets like ETTh2, ETTm1, and Weather, where Ada-MoGE maintains a consistent edge in both error metrics. Notably, Ada-MoGE's average MSE and MAE values at different time horizons are lower than those of the competing models. On the ETTh1 dataset, for instance, Ada-MoGE achieves an average MSE of 0.432, outperforming TimePro (0.438) and FreqMoE (0.440). This superior performance remains evident at longer forecasting horizons, such as 720 steps, where Ada-MoGE continues to yield more accurate predictions than models like iTransformer and PatchTST. Overall, Ada-MoGE achievs 51 first-place rankings, significantly outperforming all other models for multivariate long-term time series forecasting.

To provide a more intuitive demonstration of Ada-MoGE's forecasting performance, Fig. 3 presents a comparison of the 96-step forecasts from Ada-MoGE, FreqMoE, and TimeMixer on the ETTm2 dataset. The GroundTruth (blue) is plotted alongside the predictions (orange). While all models

Table 2: Multivariate long-term forecasting results across different horizons (H $\in \{96, 192, 336, 720\}$) under a lookback window of L = 96. Per-row best (red) and second-best (blue) results are highlighted.

| | Metric | Ada-MoGE (Ours) MSE | MAE | TimePro (ICML'25) MSE | MAE | FreqMoE (ArXiv'25) MSE | MAE | SOFTS (NeurIPS'24) MSE | MAE | TimeMixer (ICLR'24) MSE | MAE | iTransformer (ICLR'24) MSE | MAE | PatchTST (ICLR'23) MSE | MAE | TimesNet (ICLR'23) MSE | MAE |
|---|---|---|---|---|---|---|---|---|---|---|---|---|---|---|---|---|---|
| ETTh1 | 96 | 0.373 | 0.394 | 0.375 | 0.398 | 0.382 | 0.404 | 0.381 | 0.399 | 0.375 | 0.400 | 0.386 | 0.405 | 0.394 | 0.406 | 0.384 | 0.402 |
| | 192 | 0.422 | 0.426 | 0.427 | 0.429 | 0.433 | 0.429 | 0.435 | 0.431 | 0.429 | 0.421 | 0.441 | 0.436 | 0.440 | 0.435 | 0.436 | 0.429 |
| | 336 | 0.462 | 0.443 | 0.472 | 0.450 | 0.475 | 0.451 | 0.480 | 0.452 | 0.484 | 0.458 | 0.487 | 0.458 | 0.491 | 0.462 | 0.491 | 0.469 |
| | 720 | 0.469 | 0.462 | 0.476 | 0.474 | 0.485 | 0.476 | 0.499 | 0.488 | 0.498 | 0.482 | 0.503 | 0.491 | 0.487 | 0.479 | 0.521 | 0.500 |
| | Avg | 0.432 | 0.431 | 0.438 | 0.438 | 0.444 | 0.440 | 0.449 | 0.442 | 0.447 | 0.440 | 0.454 | 0.447 | 0.453 | 0.446 | 0.458 | 0.450 |
| ETTh2 | 96 | 0.287 | 0.339 | 0.293 | 0.345 | 0.290 | 0.340 | 0.297 | 0.347 | 0.294 | 0.345 | 0.297 | 0.349 | 0.288 | 0.340 | 0.340 | 0.374 |
| | 192 | 0.367 | 0.390 | 0.367 | 0.394 | 0.369 | 0.390 | 0.373 | 0.394 | 0.376 | 0.396 | 0.380 | 0.400 | 0.376 | 0.395 | 0.402 | 0.414 |
| | 336 | 0.411 | 0.429 | 0.419 | 0.431 | 0.411 | 0.427 | 0.410 | 0.426 | 0.423 | 0.436 | 0.428 | 0.432 | 0.440 | 0.451 | 0.452 | 0.452 |
| | 720 | 0.426 | 0.442 | 0.427 | 0.445 | 0.421 | 0.443 | 0.411 | 0.433 | 0.438 | 0.451 | 0.427 | 0.445 | 0.436 | 0.453 | 0.462 | 0.468 |
| | Avg | 0.373 | 0.400 | 0.377 | 0.403 | 0.373 | 0.400 | 0.373 | 0.400 | 0.383 | 0.407 | 0.383 | 0.407 | 0.385 | 0.410 | 0.414 | 0.427 |
| ETTm1 | 96 | 0.313 | 0.352 | 0.326 | 0.364 | 0.319 | 0.357 | 0.325 | 0.361 | 0.320 | 0.357 | 0.334 | 0.368 | 0.329 | 0.365 | 0.338 | 0.375 |
| | 192 | 0.358 | 0.381 | 0.367 | 0.383 | 0.363 | 0.384 | 0.375 | 0.389 | 0.361 | 0.381 | 0.377 | 0.391 | 0.380 | 0.394 | 0.374 | 0.387 |
| | 336 | 0.387 | 0.403 | 0.402 | 0.409 | 0.393 | 0.404 | 0.405 | 0.412 | 0.390 | 0.404 | 0.426 | 0.420 | 0.400 | 0.410 | 0.410 | 0.411 |
| | 720 | 0.449 | 0.439 | 0.469 | 0.446 | 0.457 | 0.443 | 0.466 | 0.447 | 0.454 | 0.441 | 0.491 | 0.459 | 0.475 | 0.453 | 0.478 | 0.450 |
| | Avg | 0.377 | 0.394 | 0.391 | 0.400 | 0.383 | 0.397 | 0.393 | 0.403 | 0.381 | 0.395 | 0.407 | 0.410 | 0.396 | 0.406 | 0.400 | 0.406 |
| ETTm2 | 96 | 0.173 | 0.256 | 0.178 | 0.260 | 0.176 | 0.259 | 0.180 | 0.261 | 0.175 | 0.258 | 0.180 | 0.264 | 0.184 | 0.264 | 0.187 | 0.267 |
| | 192 | 0.235 | 0.297 | 0.242 | 0.303 | 0.240 | 0.299 | 0.246 | 0.306 | 0.237 | 0.299 | 0.250 | 0.309 | 0.246 | 0.306 | 0.249 | 0.309 |
| | 336 | 0.292 | 0.339 | 0.303 | 0.342 | 0.299 | 0.338 | 0.319 | 0.352 | 0.298 | 0.340 | 0.311 | 0.348 | 0.308 | 0.346 | 0.321 | 0.351 |
| | 720 | 0.389 | 0.393 | 0.400 | 0.399 | 0.396 | 0.394 | 0.405 | 0.401 | 0.391 | 0.396 | 0.412 | 0.407 | 0.409 | 0.402 | 0.408 | 0.403 |
| | Avg | 0.272 | 0.321 | 0.281 | 0.326 | 0.278 | 0.323 | 0.287 | 0.330 | 0.275 | 0.323 | 0.288 | 0.332 | 0.287 | 0.330 | 0.291 | 0.333 |
| ECL | 96 | 0.153 | 0.244 | 0.139 | 0.234 | 0.152 | 0.246 | 0.143 | 0.233 | 0.153 | 0.247 | 0.148 | 0.240 | 0.164 | 0.251 | 0.168 | 0.272 |
| | 192 | 0.167 | 0.256 | 0.156 | 0.249 | 0.165 | 0.255 | 0.158 | 0.248 | 0.166 | 0.256 | 0.162 | 0.253 | 0.173 | 0.262 | 0.184 | 0.289 |
| | 336 | 0.185 | 0.275 | 0.172 | 0.267 | 0.181 | 0.274 | 0.178 | 0.269 | 0.185 | 0.277 | 0.178 | 0.269 | 0.190 | 0.279 | 0.198 | 0.300 |
| | 720 | 0.224 | 0.310 | 0.209 | 0.299 | 0.219 | 0.307 | 0.218 | 0.305 | 0.225 | 0.310 | 0.225 | 0.317 | 0.230 | 0.313 | 0.220 | 0.320 |
| | Avg | 0.182 | 0.271 | 0.169 | 0.262 | 0.179 | 0.270 | 0.174 | 0.264 | 0.182 | 0.273 | 0.178 | 0.270 | 0.189 | 0.276 | 0.192 | 0.295 |
| Weather | 96 | 0.161 | 0.209 | 0.166 | 0.207 | 0.168 | 0.215 | 0.166 | 0.208 | 0.162 | 0.209 | 0.174 | 0.214 | 0.176 | 0.217 | 0.172 | 0.220 |
| | 192 | 0.206 | 0.250 | 0.216 | 0.254 | 0.212 | 0.253 | 0.217 | 0.253 | 0.209 | 0.251 | 0.221 | 0.254 | 0.221 | 0.256 | 0.219 | 0.261 |
| | 336 | 0.261 | 0.291 | 0.273 | 0.296 | 0.268 | 0.291 | 0.282 | 0.300 | 0.265 | 0.293 | 0.278 | 0.296 | 0.275 | 0.296 | 0.280 | 0.306 |
| | 720 | 0.341 | 0.344 | 0.351 | 0.346 | 0.342 | 0.345 | 0.356 | 0.351 | 0.344 | 0.346 | 0.358 | 0.347 | 0.352 | 0.346 | 0.365 | 0.359 |
| | Avg | 0.242 | 0.273 | 0.251 | 0.276 | 0.247 | 0.276 | 0.255 | 0.278 | 0.245 | 0.275 | 0.258 | 0.278 | 0.256 | 0.279 | 0.259 | 0.287 |
| SolarEnergy | 96 | 0.182 | 0.258 | 0.196 | 0.237 | 0.221 | 0.266 | 0.200 | 0.230 | 0.189 | 0.259 | 0.203 | 0.237 | 0.205 | 0.246 | 0.250 | 0.292 |
| | 192 | 0.208 | 0.277 | 0.231 | 0.263 | 0.253 | 0.287 | 0.229 | 0.253 | 0.222 | 0.283 | 0.233 | 0.261 | 0.237 | 0.267 | 0.296 | 0.318 |
| | 336 | 0.224 | 0.280 | 0.250 | 0.281 | 0.264 | 0.296 | 0.243 | 0.269 | 0.231 | 0.292 | 0.248 | 0.273 | 0.250 | 0.276 | 0.319 | 0.330 |
| | 720 | 0.217 | 0.273 | 0.253 | 0.285 | 0.263 | 0.289 | 0.245 | 0.272 | 0.223 | 0.285 | 0.249 | 0.275 | 0.252 | 0.275 | 0.338 | 0.337 |
| | Avg | 0.208 | 0.272 | 0.232 | 0.266 | 0.250 | 0.284 | 0.229 | 0.256 | 0.216 | 0.280 | 0.233 | 0.262 | 0.236 | 0.266 | 0.301 | 0.319 |
| Average | | 0.298 | 0.337 | 0.306 | 0.339 | 0.308 | 0.341 | 0.311 | 0.342 | 0.304 | 0.341 | 0.314 | 0.344 | 0.314 | 0.345 | 0.325 | 0.353 |
| $1^{st}$ Count | | 51 | | 10 | | 4 | | 13 | | 2 | | 0 | | 0 | | 0 | |

Table 3: Contribution of individual components in Ada-MoGE to forecasting performance.

| Adaptive Learner | Gaussian Experts | ETTh1 MSE | MAE | Solar MSE | MAE |
|---|---|---|---|---|---|
| ✗ | ✗ | 0.447 | 0.44 | 0.242 | 0.296 |
| ✗ | ✓ | 0.441 ↓1.3% | 0.438 ↓0.5% | 0.229 ↓5.5% | 0.278 ↓6.3% |
| ✓ | ✓ | 0.432 ↓3.1% | 0.431 ↓2.1% | 0.208 ↓14.2% | 0.272 ↓8.1% |

are able to capture the overall trend of the data, the forecasted curves of Ada-MoGE are noticeably closer to the actual data, particularly at key points such as the peaks and valleys. In comparison, the predictions from FreqMoE and TimeMixer show more noticeable deviations from the GroundTruth, especially during the inflection points, where Ada-MoGE maintains a more accurate fit.

## 4.4 MODEL ANALYSIS

To evaluate the impact of different hyperparameters and the effectiveness of the model structure, we carry out a comprehensive set of experiments.

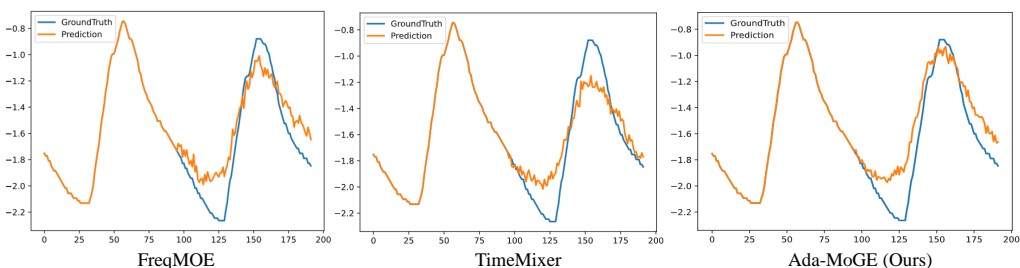

Figure 3: Comparison of 96-step Forecasts by FreqMoE, TimeMixer, and Ada-MoGE on the ETTm2 Dataset. GroundTruth (blue) versus forecasts (orange).

#### 4.4.1 ABLATION STUDY

To verify the effectiveness of the adaptive learner and the Gaussian expert, related ablation experiments are performed. As shown in Table 3, starting from the baseline without either module (ETTh1: 0.447 MSE; Solar: 0.242 MSE), adding Gaussian experts alone reduces errors (ETTh1: 0.441 MSE; Solar: 0.229 MSE), decreasing by 1.3% on ETTh1 and 5.5% on Solar. This confirms the benefit of Gaussian feature decoupling. After FFT, learnable Gaussian bandpass filters split the spectrum into compact subbands, providing each expert with a clean frequency range and reducing aliasing—especially effective for highly seasonal data like Solar. Enabling adaptive learner on top of Gaussian experts brings the largest improvements (ETTh1: 0.432 MSE; Solar: 0.208 MSE), down 3.1% and 14.2% from baseline. The gain stems from dual feature gating to activate the Top-K most relevant experts while suppressing noise-dominated bands, with the spectral decoupling of Gaussian experts.

#### 4.4.2 ANALYSIS OF DIFFERENT EXPERT NUMBER SELECTION METHODS

As shown in Table.4, with the rest of the pipeline fixed, we compare different expert number selection designs. A simple MLP gate provides modest gains over the baseline (ETTh1: 0.438 MSE, 0.432 MAE; Solar: 0.234 MSE, 0.289 MAE), though it lacks explicit spectral guidance. Squeeze-and-Excitation (SE) Attention improves variable weighting and performs better on Solar (ETTh1: 0.442 MSE, 0.436 MAE; Solar: 0.229 MSE, 0.282 MAE), yet it cannot identify dominant frequency bands. This joint frequency-aware and variable-aware gating yields the best results on both datasets (ETTh1: 0.432 MSE, 0.431 MAE; Solar: 0.208 MSE, 0.272 MAE), demonstrating that allocating expert capacity to the most predictive subbands and channels is essential.

#### 4.4.3 COMPARISON OF BETWEEN ADA-MOGE AND FREQ-MOE

The comparative results on the ETTh1 and ETTm1 datasets clearly demonstrate the advantage of integrating our proposed Ada-MoGE module over the Freq-MOE baseline. As shown in Fig. 4, Ada-MoGE consistently achieves superior performance across both TimeMixer and iTransformer models. On the ETTm1 dataset, Ada-MoGE yields a lower MSE for TimeMixer (0.377 vs. 0.383). The improvement is more pronounced on the ETTh1 dataset, where Ada-MoGE attains a lower MSE in TimeMixer (0.432 vs. 0.444). These consistent gains on key benchmarks validate that Ada-MoGE is a more effective enhancement for capturing temporal dependencies than the Freq-MOE module.

Table 4: Comparison of different expert number selection methods.

| Method | ETTh1 | | Solar | |
|---|---|---|---|---|
| | MSE | MAE | MSE | MAE |
| Baseline | 0.447 | 0.440 | 0.242 | 0.296 |
| MLP | 0.438 | 0.432 | 0.234 | 0.289 |
| SE Hu et al. (2018) | 0.442 | 0.436 | 0.229 | 0.282 |
| Dual Feature(Ours) | 0.432 | 0.431 | 0.208 | 0.272 |

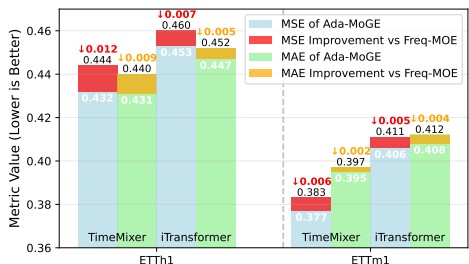

Figure 4: Performance comparison of Ada-MoGE versus Freq-MOE modules.

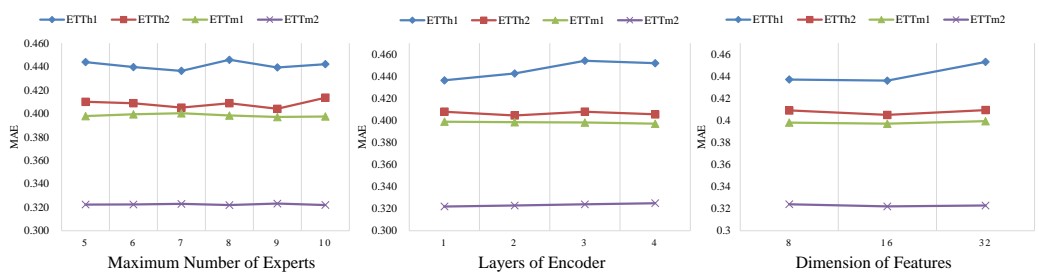

Figure 5: Hyperparameter sensitivity analysis of Ada-MoGE on ETT datasets.

### 4.4.4 ANALYSIS OF HYPERPARAMETERS

**Analysis of Maximum number of experts**  As shown in Fig. 5, when the maximum number of experts increases from 5 to 10, the model exhibits a distinct "moderate-is-best" pattern. ETTh1 achieves its lowest MAE of 0.436 with 7 experts, while ETTh2 performs best with 9 experts (MAE=0.404). ETTm1 stabilizes at its minimum error of 0.397 with 9 to 10 experts, and ETTm2 remains largely flat, with MAE between 0.322 and 0.323. These results align with the design principle of Ada-MoGE: Gaussian band-pass filtering allocates spectrally compact sub-bands to independent experts, while the dual-dimensional adaptive gating activates only the most informative bands. An insufficient number of experts prevents the model from covering the full range of dominant frequencies, leading to residual aliasing. Conversely, an excessive number of experts introduces noise-dominated sub-bands and intensifies competition within the gating mechanism, which can slightly degrade performance. Overall, a configuration of 7 to 9 experts provides the optimal balance between comprehensive frequency coverage and effective noise suppression.

**Analysis of Layers of encoder**  As shown in Fig. 5, depth brings limited gains because the frequency-domain decoupling already concentrates predictive energy into clean, narrow bands handled per expert. ETTh1 is best at 1 layer (0.436) and degrades when stacked deeper (0.454 at 3 layers, 0.452 at 4 layers). ETTh2 favors 2 layers (0.404), with deeper settings offering no improvement. ETTm1 changes marginally and is best at 4 layers (0.397), while ETTm2 is best at 1 layer (0.322) and worsens slightly as depth increases. These results suggest that 1–2 layers are generally sufficient: adding depth can re-mix already purified sub-band features, causing over-smoothing or optimization noise with little benefit.

**Analysis of Dimension of features**  As shown in Fig. 5, a feature dimension of 16 yields optimal performance, with metrics plateauing or degrading at lower (8) or higher (32) values. Specifically, this setting yields MAEs of 0.436 on ETTh1, 0.405 on ETTh2, 0.397 on ETTm1, and 0.322 on ETTm2. A moderate dimension of 16 is sufficient to encode $\mu(f)$ (Eq. 1) and $E(v)$ (Eq. 2), whereas an 8-dimensional space is too limited, causing underfitting, and a 32-dimensional space introduces redundant parameters and estimation noise that undermine gating confidence. Therefore, 16 dimensions achieves the optimal balance between parameter efficiency and model generalization.

## 5 CONCLUSION

In this paper, we propose Ada-MoGE, an adaptive Gaussian Mixture of Experts model. Ada-MoGE can effectively address the issue of frequency coverage imbalance. It integrates spectral intensity and frequency response to adaptively determine the number of experts, ensuring alignment with the input data's frequency distribution. This approach prevents both information loss due to an insufficient number of experts and noise contamination from an excess of experts. Additionally, we employ Gaussian band-pass filtering to smoothly decompose the frequency domain features to prevent noise introduction from direct band truncation. We conduct extensive experiments to validate the effectiveness of our method. The experimental results demonstrate that our approach achieves state-of-the-art performance on six benchmarks. And our method requires fewer parameters and FLOPs compared to other existing methods.

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

## A    USE OF LARGE LANGUAGE MODELS

We adopt large language models (LLMs) to aid and polish the writing of this manuscript. Specifically, LLM is used to improve grammar, wording, and clarity. However, the logic and main content of the manuscript are completed by all authors.

