# OpenReview forum: "Ada-MoGE: Adaptive Mixture of Gaussian Expert Model for Time Series Forecasting"
_ICLR.cc/2026/Conference — ICLR 2026 Conference Withdrawn Submission_

### Official Review · Reviewer_aogi · 2025-10-25

**Soundness:** 2
**Presentation:** 2
**Contribution:** 2
**Rating:** 2
**Confidence:** 4

**Summary:**

The paper proposes Ada-MoGE, a model for multivariate time series forecasting. It aims to solve the frequency coverage imbalance issue found in traditional Mixture of Experts (MoE) models, which use a fixed number of experts. The proposed model has two main components:
- An adaptive expert selection mechanism that determines the number of active experts ($K$) by fusing spectral intensity and frequency response features.
- Gaussian band-pass filters for smooth frequency feature decomposition, replacing hard truncation to reduce noise. The model is presented as having only 0.2M parameters and achieving state-of-the-art results on six public benchmarks.

**Strengths:**

- The paper addresses a valid and practical problem in time series MoE models, which is the mismatch between a fixed number of experts and the shifting spectral distributions of real world data.
- The core idea of adaptively selecting the number of experts ($K$) based on data specific spectral properties is novel and sensible. Using both frequency dominance ($\mu(f)$) and variable activity ($E(v)$) to inform this selection is a good design choice.
- The model reports very strong performance on multiple benchmarks against recent and strong baselines. Achieving this with a very small parameter count of 0.2M is impressive.

**Weaknesses:**

- The central mechanism for adaptive expert selection is not explained well. The paper says an MLP outputs the number $K$, which guides a Top K selection. It is completely unclear how this integer $K$ is trained in an end to end differentiable way.
- The paper is very confusing about the model architecture. Table 1 shows Ada MoGE as a module, but Table 2 lists "Ada MoGE (Ours)" as a separate model. The architecture of this standalone model from Table 2 is not described, which makes the main results hard to verify.
- The paper criticizes FreqMoE for "abrupt band wise truncation". This seems to misrepresent FreqMoE, which the paper itself states uses a "soft weighting approach". This weakens the motivation for the Gaussian experts.

**Questions:**

- Please explain the exact training mechanism for $K$. How do you backpropagate through the selection of an integer $K$? Is a relaxation technique or a reinforcement learning approach used?
- What is the full architecture of the "Ada MoGE (Ours)" model in Table 2? Is it a standalone model? If so, what are its components besides the expert selection and Gaussian filters?
- Does the 0.2M parameter count refer to the standalone model in Table 2 or just the parameters of the Ada MoGE module itself?

---

### Official Review · Reviewer_HruZ · 2025-10-31

**Soundness:** 2
**Presentation:** 3
**Contribution:** 2
**Rating:** 2
**Confidence:** 5

**Summary:**

In the paper under review, a mixture of experts (MoE) for time series forecasting is proposed. Key contributions are: 1) an adaptive expert selection based on the analysis of the frequency spectrum of features of the time series and 2) Gaussian experts that perform band pass filtering. The proposed MoE can be used stand-alone as well as an add-on to existing forecasting models.

**Strengths:**

* The paper is well written and easy to follow
* The proposed idea of an adaptive selection is novel and well suited for time series applications
* It is beneficial that the approach can be used in combination with existing forecasting models
* In the experiments many ablations experiments are performed to show the contributions of the individual components of the proposed approach and the sensitivity of the results to factors like numbers of experts or number of features.

**Weaknesses:**

* In the state of the are analysis the authors only focus on machine learning and here, on neural networks. Other, more classical, approaches are completely ignored.

* Although being novel, the proposed contribution is rather incremental. The performance gains by the proposed MoE addon are given but rather limited. I considere it too little for a major ML conference like the ICLR.

* There should be one paragraph in the paper that describes in more detail, how the proposed approach is combined with existing methods. Currently, this description is too limited and often implicit.

* The descriptions of the experiments lacks some details. It is not mentioned for the results of Table 2, the Ada-MoGE is combined with TimeMixture. Merely by comparing the numbers of Table 1 with Table 2 the reader can deduce that Table 2 is based on the aforementioned combination.

* Although the authors motivate time forecasting with application from manufactuing, among others, no typical time series from the manufacturing domain is used in the experiments. For time series in manufacturing it is vrey common that they are of high frequency (1kHZ and more). The used data sets howevere, are only low frequency. Thus, it is not clear if and how the proposed approach can be applied in such scenarios.

**Questions:**

Why is band bass filtering based on Gaussian functions? Isn't there the risk that if $f_1$ and $f_2$ are far apart also frequencies between $f_1$ and $f_2$ are surpressed? Why not using a rectangular-like band pass?

**Details Of Ethics Concerns:**

There are no ethics concerns.

---

### Official Review · Reviewer_PCe2 · 2025-10-31

**Soundness:** 3
**Presentation:** 3
**Contribution:** 2
**Rating:** 4
**Confidence:** 3

**Summary:**

The paper introduces an adaptive frequency-domain mixture-of-experts framework for time-series forecasting, dynamically selecting the number of experts based on spectral statistics. This formulation is intuitive and potentially valuable, as dominant frequency components can vary across samples, making fixed expert assignment suboptimal. The use of learnable Gaussian filters for smooth band decomposition provides a differentiable alternative to hard frequency partitioning and adds modeling flexibility. That said, while the core idea is promising and empirically validated, additional clarity on certain methodological aspects and situating the contributions more fully within recent frequency-aware literature would further strengthen the work.

**Strengths:**

1. The paper introduces an intuitive adaptive frequency-domain MoE framework that dynamically selects experts based on spectral characteristics.

2. The use of learnable Gaussian filters provides smooth and flexible frequency decomposition beyond hard band partitioning.


3. The paper is clearly written with a coherent motivation and strong empirical results.

**Weaknesses:**

1. The motivation for operating in the frequency domain remains insufficiently justified. While the paper argues that dominant frequencies vary, it does not clearly explain why frequency-domain routing is fundamentally preferable to potential time-domain or variable-domain alternatives (e.g., learnable filters, SSMs, or channel-wise gating).

2. The experimental comparisons do not fully place the method within the broader literature on frequency-aware prediction. Other representative frequency-based architectures, such as [1] and [2], are not included in the main results, making it difficult to assess gains relative to frequency modeling approaches rather than only MoE variants.

3. Some methodological details are not fully clarified, which may affect reproducibility. For example, the construction of the gated input function, the precise routing mechanism for determining $K$, constraints on the learnable frequency boundaries, and the definition of terms such as $D_j$ are not clearly described.

[1] Xu, Zhijian, et al. “FITS: Modeling Time Series with 10k Parameters.” International Conference on Learning Representations (ICLR), 2024.

[2] Zhang, et al. “Not All Frequencies Are Created Equal: Towards a Dynamic Fusion of Frequencies in Time-Series Forecasting.” ACM Multimedia (ACM MM), 2024.

**Questions:**

1. The paper reports parameters and FLOPs, but could you also provide practical training metrics (e.g., actual wall-clock time, memory usage, convergence behavior, throughput)? These measurements are important to substantiate the claimed efficiency.

2. The validation of the adaptive expert counting mechanism appears insufficient. Beyond ablation removal, is there empirical evidence that the learned number of experts reflects spectral complexity or frequency-structure characteristics?

3. Relatedly, can you show whether the learned expert count correlates with dominant-frequency patterns (e.g., number of spectral peaks, spectral entropy, or energy concentration)? Such analysis would concretely support the adaptivity motivation.

---

### Official Review · Reviewer_QTa3 · 2025-11-03

**Soundness:** 2
**Presentation:** 1
**Contribution:** 2
**Rating:** 2
**Confidence:** 4

**Summary:**

This paper explores the use of frequency information from multivariate time series to guide expert selection in mixture-of-experts architectures. The proposed approach is compatible with certain existing multivariate time series models and is evaluated empirically.

However, the presentation requires substantial revision before the paper can reach a publication-ready standard. This paper currently contains redundant descriptions, unclear terminology, and loosely defined concepts, which make it difficult to follow and obscure the main contributions, as elaborated in the weaknesses section.

**Strengths:**

1. This paper is motivated by the observation that existing mixture-of-experts models may suffer from a frequency imbalance issue. While this is an interesting and relevant motivation, the proposed method does not appear to address the problem in a clear or rigorous manner.

2. The proposed approach seems to be compatible with existing methods, which allows for evaluation in terms of the incremental benefits it provides over the chosen base model.

**Weaknesses:**

1. Claims lacking proper justification

In the abstract and introduction, the authors discuss a frequency shift phenomenon in time series data and claim that “Traditional Mixture of Experts (MoE) models, which employ a fixed number of experts, struggle to adapt to these changes, resulting in a frequency coverage imbalance issue.” However, the connection between this phenomenon and the limitations of MoE models is not clearly established. A more detailed explanation—ideally supported by theoretical reasoning, empirical evidence, or illustrative observations—is needed to substantiate this claim.

In Section 3.1, the statement that “using a fixed number of experts often leads to an imbalance in frequency domain coverage” also lacks sufficient justification or explanation.

2. Use of confusing terminology

In Section 3.1, the term “frequency response” is used ambiguously and should be clarified.

It is also unclear how the proposed method achieves the goal of “ensuring that the number of experts matches the frequency distribution of the input data,” given that Equations (1)–(5) involve only simple formulations that do not explicitly reflect such a mechanism.

Similarly, the claim that “this approach avoids information loss due to too few experts and noise introduction due to too many experts” is confusing and lacks adequate justification.

3. Equations (1)–(3) in Section 3.2 merely aggregate frequency and variable-wise data before feeding the concatenated representation into a dense layer. This procedure appears straightforward and does not seem directly related to the stated objective of addressing frequency balancing and covering in the Mixture-of-Experts framework.

4. In the experiments, the model still requires setting a hyperparameter for the maximum number of experts. This raises doubts about the claimed contribution, as it suggests that the proposed method does not fully eliminate the need for manual specification, which was one of the motivations.

**Questions:**

Refer to the weaknesses section.

---

### Note · Authors · 2025-11-27

I have read and agree with the venue's withdrawal policy on behalf of myself and my co-authors.